# Tooth Loss, Nutrition, and Oral Health-Related Quality of Life in Older Adults: Evidence from a Structural Equation Model

**DOI:** 10.3390/ijerph22121793

**Published:** 2025-11-26

**Authors:** Beatriz Della Terra Mouco Garrido, Patrícia Soares Silva Pereira, Kamilly Foloni, Heloisa Dalberto Pegoraro, Isabelle Roldão Souza, José Roberto Magalhães Bastos, Gerson Aparecido Foratori-Júnior, Monica Yamauti, Raquel Conceição Ferreira, Roosevelt Silva Bastos

**Affiliations:** 1Department of Pediatric Dentistry, Orthodontics, and Public Health, Universidade de São Paulo, Bauru 17012-901, Brazil; beatriz.garrido@usp.br (B.D.T.M.G.); patriciasoares.ssp@gmail.com (P.S.S.P.); kamilly.foloni@gmail.com (K.F.); heloisadpegoraro@gmail.com (H.D.P.); isabelleroldao@usp.br (I.R.S.); zeromaba@fob.usp.br (J.R.M.B.); gerson.foratori@usp.br (G.A.F.-J.); 2Department of Restorative Dentistry, Graduate School of Dental Medicine, Hokkaido University, Sapporo 060-8586, Japan; myamauti@den.hokudai.ac.jp; 3Department of Community and Preventive Dentistry, Federal University of Minas Gerais, Belo Horizonte 31270-901, Brazil; raquelcf@odonto.ufmg.br

**Keywords:** tooth loss, oral health-related quality of life, nutrition, older adults, structural equation modeling

## Abstract

Background: Oral health problems, particularly tooth loss, may impair nutrition and reduce oral health-related quality of life (OHRQoL) in older adults. However, the pathways linking these conditions remain unclear. Methods: We analyzed data from 112 older adults (≥60 years). We measured OHRQoL using the Geriatric Oral Health Assessment Index (GOHAI) and nutritional status using the Mini Nutritional Assessment (MNA), and used tooth loss as a clinical indicator. Bivariate analyses used Spearman’s correlation, and structural equation modeling (SEM) with robust estimation was applied to assess direct and indirect pathways. Mediation was evaluated with bootstrap resampling. Results: The prevalence of impaired OHRQoL (GOHAI > 0) was 25.9% (95% CI: 18.6 to 34.9). Tooth loss correlated with worse nutrition (ρ = −0.32; 95% CI: −0.48 to −0.15), and poorer nutrition was associated with worse OHRQoL (ρ = −0.22; 95% CI: −0.40 to −0.03). SEM showed that tooth loss negatively affected nutrition (β = −0.21; 95% CI: −0.43 to 0.00) and that nutrition was directly associated with OHRQoL (β = −0.21; 95% CI: −0.34 to −0.08). Bootstrap analyses confirmed a direct effect of tooth loss on OHRQoL (β = 0.19; 95% CI: 0.02 to 0.36), while the indirect pathway through nutrition did not reach statistical significance. Conclusions: Tooth loss and poor nutrition independently reduce OHRQoL in older adults. Although mediation by nutrition was not statistically significant, the findings highlight the interdependence of oral and general health and support integrated public health strategies for aging populations.

## 1. Introduction

Oral health is an essential component of overall health and well-being. Tooth loss is highly prevalent in this population and represents both a marker of cumulative oral disease and a determinant of functional capacity, affecting chewing ability, dietary habits, social participation, and quality of life [1,2,3,4]. Oral health–related quality of life (OHRQoL), commonly assessed using instruments such as the Geriatric Oral Health Assessment Index (GOHAI), reflects the impact of oral conditions on daily activities, psychological well-being, and social relationships [2,4]. The GOHAI was selected for this study because it was specifically developed to evaluate OHRQoL in older adults, emphasizing functional and psychosocial dimensions that are highly relevant in this population [5]. The instrument has also been validated for use in Brazil [6] and has demonstrated comparable results to the OHIP-14 when applied among older adults, further supporting its appropriateness for this research [7]. A reduced number of natural teeth or edentulism is strongly associated with functional impairments, altered nutrition, and poor overall health [1,3,8,9]. Inadequate nutrition has also been linked to frailty, systemic diseases, and reduced life expectancy [10,11]. Moreover, compromised OHRQoL is more frequent among older individuals with tooth loss and poor nutritional status, reinforcing the interdependence of oral and general health [4,8]. Collectively, these findings highlight that the consequences of tooth loss extend beyond the oral cavity, influencing nutrition, systemic conditions, and overall well-being in later life. In this study, older adults are defined as individuals aged 60 years and over according to the World Health Organization’s classification for developing countries.

Although the links between tooth loss, nutritional status, and quality of life are well documented, important gaps remain in the literature. Most studies have examined these relationships separately, showing that tooth loss compromises mastication and dietary intake, and that both conditions are related to poorer OHRQoL [8,11]. However, few investigations have explored whether nutritional status may act as a mediator in the pathway between tooth loss and quality of life. Evidence from Brazil using structural equation modeling suggested that tooth loss influences protein intake and OHRQoL [12], but this approach remains rare and limited in scope. Moreover, most available evidence comes from high-income countries, with relatively few studies conducted in Latin America despite the high burden of oral diseases and social inequalities in the region [1]. In addition, although socioeconomic disparities in oral health and quality of life are consistently reported [2,13], they are seldom integrated with nutritional factors in analytical models. These gaps emphasize the need for studies in middle-income settings, using robust statistical approaches, to clarify whether and to what extent nutrition mediates the association between tooth loss and OHRQoL.

In Brazil, this focus is especially important because approximately 68% of the population lives in municipalities with 40,000 inhabitants or fewer, where health services are more limited and inequalities in access to dental care are persistent. In these contexts, primary health care is widely implemented through the Family Health Strategy, but access to specialized oral health services remains a challenge. Given the growing burden of oral diseases and the demographic transition in Brazil, where the proportion of older adults is rapidly increasing, investigating the pathways linking oral health, nutrition, and quality of life is of particular relevance. The World Health Organization and international consensus reports have emphasized that maintaining natural dentition is fundamental for healthy ageing, since tooth loss affects nutrition, systemic conditions, and overall well-being [14]. Evidence from scoping reviews also highlights tooth retention as a key determinant of quality of life in older populations worldwide [1]. However, few studies in middle-income countries have applied advanced methods to explore these complex relationships. Therefore, this study aimed to assess the associations among tooth loss, nutritional status, and OHRQoL in older adults, and to evaluate the potential mediating role of nutrition in this relationship.

## 2. Methods

### 2.1. Study Design and Participants

This cross-sectional, population-based study was conducted to investigate the relationship between tooth loss, nutritional status, and oral health-related quality of life (OHRQoL) in older adults. Data were collected from 112 individuals aged 60 years or older, residing in the municipality of Boraceia, São Paulo State, Brazil, between April and June 2025. Boraceia is a small municipality with fewer than 5000 inhabitants (4838 in 2025 according to the Brazilian Institute of Geography and Statistics—IBGE). Although this recruitment strategy may limit external validity, it was selected for feasibility and ethical reasons, ensuring the inclusion of community-dwelling older adults engaged in the public primary health care system. The municipality of Boraceia was chosen because it exemplifies small Brazilian towns (fewer than 40,000 inhabitants) that together account for approximately 68% of the national population. These municipalities share comparable sociodemographic characteristics and are organized primarily through the Family Health Strategy. In Boraceia, 100% of the population had primary health care coverage, which enhances the reliability of the findings for local health service management and provides a model that can inform similar small municipalities across Brazil. The inclusion criteria were age ≥ 60 years, residence in the municipality for at least one year, and agreement to participate and provided informed consent. Cognitive status was evaluated using the Mini-Mental State Examination (MMSE; known in Portuguese as Mini-Exame do Estado Mental—MEEM), a brief and widely validated screening instrument for global cognitive function [15,16]. The test was applied individually by trained researchers before data collection, and cutoff scores were adjusted according to education level: <20 points for illiterate participants; 25 points for 1–4 years of schooling; 26.5 for 5–8 years; 28 for 9–11 years; and 29 for more than 11 years [15]. Despite not assessing all cognitive domains, the MMSE/MEEM remains the most frequently used and best-studied cognitive screening tool in Brazil and internationally [16]. Participants who did not reach the minimum score were excluded from the study. Of the 200 older adults initially screened, 50 were excluded due to cognitive impairment and 38 for incomplete data, resulting in a final analytic sample of 112 participants. This strategy ensured the inclusion of community-dwelling older adults capable of reliably answering the questionnaires, while capturing the specific post-pandemic context of oral and general health in a socially vulnerable setting.

### 2.2. Variables

The study included outcome, exposure, mediator, and sociodemographic variables, which were analyzed both in their original scales and in categorized formats for descriptive purposes.

The outcome was oral health-related quality of life (OHRQoL), assessed with the Geriatric Oral Health Assessment Index (GOHAI), an instrument widely applied in geriatric populations to capture the functional, psychological, and social impacts of oral health conditions [5,6]. For descriptive purposes, GOHAI scores (0–24, with higher values indicating worse OHRQoL) were summarized using means (standard deviations) and medians (interquartile ranges). The exposure was tooth loss, clinically assessed using the Caries Assessment Spectrum and Treatment (CAST) index. Teeth classified as missing due to dental caries (CAST code 8) or other reasons (CAST code 9) were included in the count of missing teeth, whereas unerupted or congenitally absent teeth (CAST code 0) were excluded. For descriptive presentation, tooth loss was categorized as <20 or ≥20 missing teeth, following the criterion that at least 20 natural teeth are required to maintain functional dentition and adequate oral function in older adults [1,14,17]. The mediator was nutritional status, measured using the Mini Nutritional Assessment (MNA), a validated tool for older adults that includes 18 items addressing anthropometric, dietary, and global assessments [11]. MNA total scores range from 0 to 30 and were categorized for descriptive purposes as: normal (≥24), at risk of malnutrition (17–23.5), and malnourished (<17). Covariates included sociodemographic characteristics: age group (60–74; ≥75 years), sex (female, male), family income (<3 minimum wages; ≥3), and educational attainment (<5 years; ≥5).

### 2.3. Statistical Analysis

The data analysis was designed to describe the study population and explore the interrelations among oral health, nutrition, and quality of life (Figure 1).

Initially, descriptive statistics summarized the study variables using measures of central tendency and dispersion for continuous variables, and frequencies with 95% confidence intervals for categorical variables. The prevalence of impaired OHRQoL (GOHAI > 0) was estimated across sociodemographic and clinical groups. Correlations between continuous variables (GOHAI score, MNA score, and number of missing teeth) were examined using Spearman’s coefficients [18].

To assess the hypothesized pathways among the study variables, we used structural equation modeling (SEM) with robust maximum likelihood estimation, which allows for simultaneous testing of direct and indirect effects while accounting for measurement errors [12]. SEM was chosen because it provides a comprehensive framework to evaluate whether the impact of tooth loss on OHRQoL occurs directly or is mediated through nutritional status. Continuous variables (GOHAI, MNA, number of missing teeth, and age) were standardized as z-scores to facilitate interpretation and comparability of coefficients. The model was adjusted for age, sex, family income, and educational attainment. Indirect effects and the proportion mediated were estimated using bootstrap resampling with 1000 replications, a recommended strategy to increase the precision of confidence intervals in mediation analysis and to provide robust estimates regardless of data normality [12]. Model fit was assessed using likelihood-based and comparative indices. Because the recursive path model was just identified (df = 0), global fit indices (CFI, TLI, RMSEA) are not informative; therefore, inference focused on standardized path coefficients and bootstrap confidence intervals. For completeness, the baseline comparison yielded a CFI = 1.000, TLI = 1.000 and RMSEA = 0.000 (pclose = 1.000), while SRMR was not reported due to missingness under FIML. Model fit was assessed using comparative indices (CFI, TLI, RMSEA), all of which indicated excellent fits.

All analyses were performed in Stata 18.0 (StataCorp, College Station, TX, USA), and a 5% significance level was adopted to guide interpretation.

### 2.4. Ethics

The study protocol was approved by the Brazilian National Research Ethics Commission (CONEP) under CAAE 79324524.6.0000.5417, in accordance with national regulations (Resolution CNS nº 510/2016) and the principles of the Declaration of Helsinki (2013). All participants were informed about the objectives and procedures of the study and provided written informed consent prior to enrollment, ensuring voluntary participation, confidentiality, and ethical compliance.

## 3. Results

A total of 112 older adults were included in the analysis, all with complete data available for sociodemographic, clinical, and nutritional variables. The mean (SD) GOHAI total score was 3.15 (3.36), with a median of 2 (IQR: 0–4.5; range 0–16), while the mean (SD) MNA score was 11.97 (2.17), with a median of 12 (IQR: 11–14; range 6–14). The prevalence of impaired oral health–related quality of life (GOHAI > 0) was higher among participants aged 60–74 years (28.77%) compared with those aged ≥75 years (20.51%). Similar prevalences were observed between women (25.84%) and men (26.09%). Socioeconomic differences were evident: impaired OHRQoL was more frequent among individuals with a family income below three minimum wages (28.07%) and among those with lower educational attainment (30.9%) compared to those with higher schooling (21.05%). Clinically, tooth loss showed marked contrasts, with participants presenting ≥20 missing teeth having almost twice the prevalence of impaired OHRQoL (37.50%) compared with those with fewer than 20 missing teeth (19.4%). According to nutritional status, the prevalence was 29.17% among well-nourished individuals, 20.59% among those at risk of malnutrition, and 16.67% among malnourished participants, although the confidence intervals were wide in the latter group. Overall, one in four older adults in the study population (25.89%; 95% CI: 18.60–34.90) reported impaired OHRQoL (Table 1).

Correlation analyses were performed to examine the relationships among oral health-related quality of life, nutritional status, and tooth loss (Table 2). A greater number of missing teeth was significantly associated with lower MNA scores (r = −0.32; 95% CI: −0.48 to −0.15), and poorer nutritional status was correlated with worse OHRQoL (r = −0.22; 95% CI: −0.40 to −0.03). In contrast, no significant correlation was observed between tooth loss and GOHAI scores (r = −0.03; 95% CI: −0.20 to 0.15).

The structural equation model (SEM) confirmed significant direct effects of tooth loss and nutritional status on OHRQoL (Table 3 and Figure 2). Tooth loss was negatively associated with nutritional status (β = −0.21; 95% CI: −0.43 to 0.00), and poorer nutritional status was associated with worse OHRQoL (β = −0.21; 95% CI: −0.34 to −0.08). In addition, tooth loss showed a direct positive effect on impaired OHRQoL (β = 0.19; 95% CI: 0.03 to 0.35). Bootstrap mediation analysis evaluated the role of nutritional status in the association between tooth loss and OHRQoL. The direct effect of tooth loss on impaired OHRQoL was significant (β = 0.19; 95% CI: 0.02 to 0.36). The indirect effect through nutritional status was not statistically significant (β = 0.04; 95% CI: −0.01 to 0.12). The proportion mediated was 0.19 (95% CI: −0.01 to 0.75), indicating a non-significant mediation pathway (Table 4).

The recursive path model was just identified (df = 0); therefore, global fit indices (CFI, TLI, RMSEA) are not informative. Interpretation focused on standardized path coefficients and bootstrap confidence intervals. Bootstrap mediation analysis (requested 5000 replications; 3199 completed) using percentile 95% CIs indicated a non-significant indirect effect (β = 0.05; 95% CI: −0.02 to 0.13) and non-significant direct (β = 0.12; −0.15 to 0.36) and total effects (β = 0.17; −0.10 to 0.42). The proportion mediated was unstable and not statistically significant.

## 4. Discussion

Tooth loss and poor nutritional status were associated with impaired oral health-related quality of life (OHRQoL) in older adults, reinforcing previous evidence that the number of remaining teeth is a critical marker of functional capacity and well-being [1,3,18]. Participants with 20 or more missing teeth reported a substantially higher prevalence of impaired OHRQoL, and lower MNA scores were significantly correlated with poorer GOHAI scores. Structural equation modeling indicated that tooth loss was directly related to worse nutritional status and OHRQoL, while the hypothesized indirect pathway through nutrition did not reach statistical significance. Interestingly, a positive direct effect of tooth loss on OHRQoL was also observed in the model, possibly reflecting adaptive psychosocial mechanisms among older adults who, despite severe tooth loss, maintain satisfaction with their oral condition through prosthetic use or social acceptance of tooth loss as part of aging. These results contribute to the literature by demonstrating, in a Brazilian community-based sample, that the influence of tooth loss on OHRQoL is both direct and partially explained by nutritional status, although the mediation was not statistically significant [12].

A reduced number of teeth or edentulism negatively influences chewing ability and dietary choices, with direct consequences for nutritional status and OHRQoL [3,8]. Studies conducted in Brazil and other countries have also demonstrated that tooth loss is related to inadequate food intake and nutritional deficits, reinforcing its systemic impact [9,12]. In institutionalized populations, poor dentition and reduced occlusal pairs were associated with lower masticatory efficiency, decreased nutrient consumption, and diminished quality of life [8,19]. Furthermore, our study identified that older adults with higher education and income tended to have better nutritional status and OHRQoL, suggesting that socioeconomic resources may buffer the negative effects of tooth loss through improved access to dental care, healthier diets, and prosthetic rehabilitation. Moreover, population-based research in the United States highlighted persistent socioeconomic disparities, with older adults of lower income or minority groups experiencing greater tooth loss and worse oral health outcomes [2,13]. These findings corroborate our results within a consistent international and national literature, and they highlight the importance of further exploring the pathways—particularly the potential mediating role of nutrition—that may explain how tooth loss compromises quality of life in older populations.

Interestingly, the model revealed a small positive direct effect of tooth loss on OHRQoL, a finding that may appear paradoxical but has been reported in other studies exploring subjective well-being in older adults [1,3,11,20]. This result may reflect an adaptive process in which older individuals, particularly those using dentures or living in social contexts where edentulism is culturally normalized, adjust their expectations and self-perceptions of oral health. Such psychosocial adaptation, combined with successful prosthetic rehabilitation, can reduce the perceived impact of tooth loss and even lead to higher satisfaction with daily oral function [21,22,23]. Moreover, this response is consistent with the concept of “response shift,” where individuals recalibrate their internal standards of quality of life as they age and experience chronic conditions [23,24]. Therefore, the positive coefficient does not imply that tooth loss improves OHRQoL, but rather that older adults may reinterpret or compensate for its consequences, maintaining subjective well-being despite objective impairment.

The mechanisms underlying the associations observed in this study can be explained by the central role of teeth in maintaining adequate nutrition and social functioning. Severe tooth loss limits masticatory capacity, particularly for fibrous and protein-rich foods, leading to changes in dietary patterns and nutritional deficits that negatively affect physical health and well-being [8,19]. Qualitative research has also shown that older adults with tooth loss adopt both adaptive strategies (such as food modification) and maladaptive behaviors (such as food avoidance and social withdrawal), which directly compromise eating experience and OHRQoL [21]. In addition, prosthetic rehabilitation can partially mitigate these effects: studies indicate that complete or implant-supported dentures improve chewing ability and quality of life, although improvements in nutritional intake are less consistent [22,23]. Although the indirect effect of nutrition was not statistically significant in our analysis, these findings support the hypothesis that diet may act as a biological and behavioral pathway linking tooth loss to impaired OHRQoL.

A major strength of this study lies in the combined use of validated instruments for oral health-related quality of life (GOHAI) and nutritional status (MNA), along with clinical assessment of tooth loss and the application of robust statistical approaches such as structural equation modeling (SEM) and bootstrap resampling. This methodological strategy allows a more comprehensive exploration of direct and indirect relationships, representing an advance over most previous research that relied solely on bivariate analyses [12]. Nevertheless, some limitations should be considered. The cross-sectional design restricts causal inference, preventing conclusions about the temporal sequence between tooth loss, nutrition, and OHRQoL. The relatively small sample size may have reduced statistical power to detect mediation effects with precision. In addition, although the MNA is a validated tool for screening nutritional status [11], it does not provide detailed information on food consumption patterns, which could have limited the assessment of diet as a mediator. Finally, the study was conducted in a single Brazilian municipality in the post-COVID-19 period, when the pandemic’s socioeconomic and health repercussions could still influence both dietary conditions and quality of life, which should be considered when interpreting the findings [14,25]. Despite these limitations, the use of population-based data, standardized oral health indices, and advanced analytical models strengthens the evidence produced and supports the reliability of our findings.

The present findings reinforce the need for a broader and more integrated approach to oral health care in aging populations. International consensus highlights that maintaining natural dentition is a cornerstone of healthy ageing, as oral conditions directly affect nutrition, systemic health, and quality of life [14]. In addition, socioeconomic disparities remain key determinants of both tooth loss and OHRQoL, as demonstrated in studies with older adults in the United States [2,13]. These inequities are particularly relevant in middle-income countries such as Brazil, where structural barriers to health care and social vulnerabilities persist [26]. Therefore, strategies that combine oral health promotion, prosthetic rehabilitation, and nutritional assessment within primary health services are essential to reduce inequalities and to improve the overall well-being of older adults.

## 5. Conclusions

Tooth loss and poor nutritional status were associated with impaired oral health-related quality of life among older adults. Although nutritional status did not significantly mediate the relationship, the findings reinforce the interconnection between oral and general health and highlight the need for integrated strategies in primary care to promote healthy aging.

## Figures and Tables

**Figure 1 ijerph-22-01793-f001:**
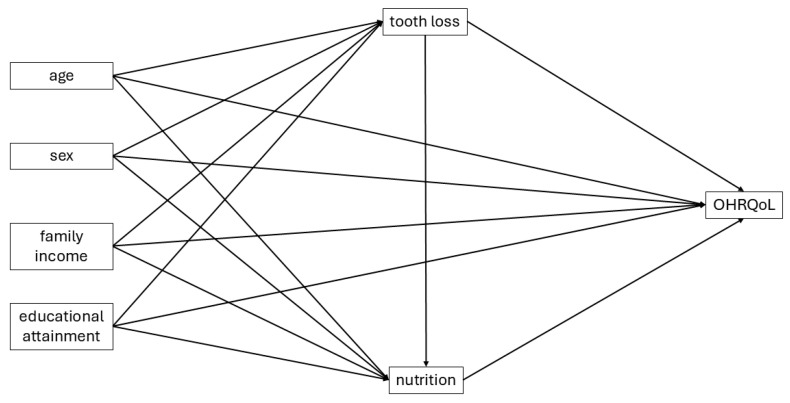
Theoretical model of the relationships among tooth loss, nutritional status, and oral health-related quality of life in older adults.

**Figure 2 ijerph-22-01793-f002:**
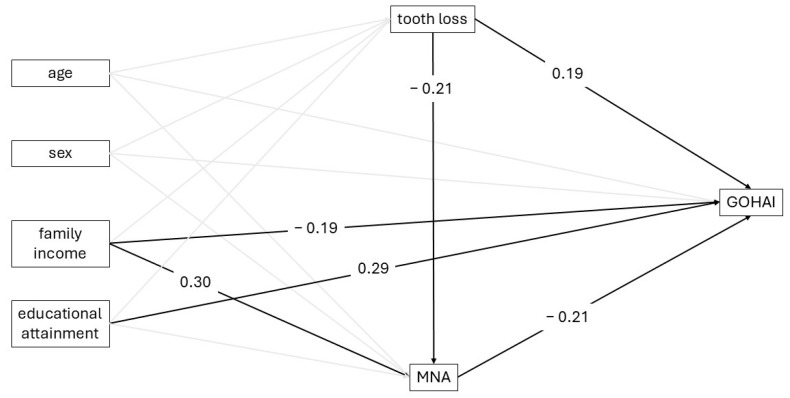
Structural equation model (SEM) of the associations among tooth loss, nutrition, and oral health-related quality of life (GOHAI) in older adults.

**Table 1 ijerph-22-01793-t001:** Prevalence of low oral health-related quality of life (GOHAI > 0) and 95% confidence intervals according to sociodemographic, clinical and nutritional characteristics.

	GOHAI > 0	
	%	IC95%
Age		
60–74	28.77	19.50–40.30
75+	20.51	10.50–36.20
Sex		
female	25.84	17.70–36.00
male	26.09	12.10–47.50
Minimum family income (MW)		
<3	28.07	17.90–4.12
3+	23.64	14.20–23.67
Educational attainment (in years)		
≥5	21.05	12.30–33.70
<5	30.91	20.10–44.40
tooth loss		
<20	19.44	11.80–30.30
≥20	37.50	23.90–53.40
Nutrition		
normal	29.17	19.80–40.80
at risk	20.59	10.10–37.50
malnourished	16.67	2.20–63.70
Total	25.89	18.60–34.90

Note: GOHAI = Geriatric Oral Health Assessment Index, an instrument used to assess oral health-related quality of life in older adults (scores > 0 indicate impaired OHRQoL).

**Table 2 ijerph-22-01793-t002:** Spearman’s correlation coefficients between oral health-related quality of life, nutritional status, and tooth loss.

	r	*p*
GOHAI × MNA	−0.22	0.023
tooth loss × MNA	−0.32	0.001
GOHAI × tooth loss	−0.03	0.772

**Table 3 ijerph-22-01793-t003:** Structural equation model (SEM): standardized coefficients, 95% confidence intervals, and *p*-values for predictors of nutritional status (MNA) and oral health-related quality of life (GOHAI).

Outcome	Predictor	Std. Beta	CI95%		*p*
MNA	tooth loss	−0.21	−0.43	0.00	0.046
	age	−0.16	−0.33	0.01	0.074
	sex	0.10	−0.05	0.26	0.194
	family income	0.30	0.13	0.46	0.001
	educational attainment	−0.10	−0.32	0.13	0.398
	constant	−0.22	−0.55	0.12	0.201
GOHAI	MNA	−0.21	−0.34	−0.08	0.002
	tooth loss	0.19	0.03	0.35	0.023
	age	−0.08	−0.22	0.05	0.235
	sex	−0.08	−0.17	0.02	0.116
	family income	−0.19	−0.35	−0.04	0.016
	educational attainment	0.29	0.16	0.42	0.001
	constant	0.40	0.13	0.67	0.003

Note: Reference groups: sex = female; age = 60–74 years; family income ≥ 3 minimum wages; educational attainment ≥ 5 years.

**Table 4 ijerph-22-01793-t004:** Bootstrap mediation analysis: direct, indirect, and total effects of tooth loss on oral health-related quality of life (GOHAI) mediated by nutritional status (MNA).

Effect	Estimate	CI95%		*p*
tooth loss → GOHAI (direct)	0.19	0.02	0.36	0.030
tooth loss → GOHAI (indirect by MNA)	0.04	−0.01	0.12	0.151
tooth loss → GOHAI (total)	0.23	0.07	0.41	0.006
Proportion mediated (PM)	0.19	−0.01	0.75	0.713

## Data Availability

Data available on request due to restrictions (Data from this component of the study may be made available upon reasonable request to the corresponding author and with approval from the Research Ethics Committee).

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
