# Peer review of "Tooth Loss, Nutrition, and Oral Health-Related Quality of Life in Older Adults: Evidence from a Structural Equation Model"

_ijerph, 2025, doi:10.3390/ijerph22121793_

Round 1

Reviewer 1 Report

Comments and Suggestions for Authors

The study was conducted to investigate the relationship between oral health-related quality of life and oral health problems in older adults. Numerous studies have been conducted relating quality of life to many possible factors.

The methodology used is standard among similar studies. The statistical analysis is also well thought out. A small sample size could limit the generalizability of the finding. 

The finding is not unexpected, since numerous studies have shown similar results. 

The study is important because it adds to another layer of scientific exploration of the relationship between older adults' quality of life and their oral health.

Author Response

We would like to thank the Editors and Reviewers for their careful reading of our manuscript and for the insightful comments and constructive suggestions that have helped us to substantially improve the clarity, coherence, and scientific rigor of the paper. All comments were carefully addressed, and corresponding revisions were made throughout the manuscript. Below, we provide a detailed, point-by-point response, indicating the changes implemented and the specific sections or line numbers where each modification appears. Text that was included or revised in the manuscript is presented in italics within quotation marks for clarity.

Reviewer 1

Comments 1:

The study was conducted to investigate the relationship between oral health–related quality of life and oral health problems in older adults. Numerous studies have been conducted relating quality of life to many possible factors.

Response 1:

We thank the reviewer for this observation. While several studies have examined the association between oral health and quality of life in older adults, our study contributes additional evidence by applying a structural equation modeling (SEM) approach to explore both direct and indirect effects through nutritional status. Furthermore, the analysis was conducted in a Brazilian small-city context, where most older adults depend on public primary health care, a population often underrepresented in the literature. This combination of analytical depth and contextual focus provides a novel perspective within the broader field of oral health–related quality of life research.

Comments 2:

The methodology used is standard among similar studies. The statistical analysis is also well thought out. A small sample size could limit the generalizability of the findings.

Response 2:

We appreciate the reviewer’s positive evaluation of our methodological and analytical approach. We agree that the relatively small sample size may limit the generalizability of the findings. This limitation has been explicitly acknowledged in the Discussion section (paragraph 4), where we emphasize that, despite this constraint, the use of validated instruments and robust statistical modeling (SEM with bootstrap resampling) supports the reliability of the observed associations.

Comments 3:

The finding is not unexpected, since numerous studies have shown similar results.

Response 3:

We thank the reviewer for this observation. Indeed, our findings are consistent with previous research demonstrating the strong link between tooth loss, nutrition, and oral health–related quality of life in older adults. This consistency reinforces the validity of our results and contributes to the cumulative evidence base on this topic. However, our study adds to existing knowledge by applying structural equation modeling (SEM) to disentangle direct and indirect pathways and by focusing on a community-dwelling older population in a small Brazilian municipality, a context rarely explored in previous studies.

Comments 4:

The study is important because it adds another layer of scientific exploration of the relationship between older adults' quality of life and their oral health.

Response 4:

We sincerely thank the reviewer for recognizing the relevance of our study. Our intention was precisely to contribute an additional analytical perspective to this field by examining how tooth loss, nutrition, and oral health–related quality of life interact within the framework of healthy ageing. We believe that the use of validated instruments and advanced analytical modeling (SEM) strengthens the scientific understanding of these relationships and supports future strategies aimed at improving the oral and general health of older adults.

Reviewer 2 Report

Comments and Suggestions for Authors

The study is interesting and well written. However, I have some commnets.

The introduction is rich in content, but there are repetitive phrases about tooth loss, nutrition, and OHRQoL that don’t really add new information. I suggest condensing this into one paragraph to improve clarity. Also, the definition of “older adult” should be clearly stated in the introduction. Furthermore, the study was conducted in a small city in Brazil, which needs motivation.

Line 26:
I’m wondering if the original reference actually supports the phrase “particularly in older adults.” The sentence reads:

Oral health is an essential component of overall health and well-being, particularly  in older adults.

Line 29

This part could be developed a bit more. While GOHAI is mentioned, the most common OHRQoL index is actually OHIP-14 within research. I suggest explaining why GOHAI was chosen for this study, along with a relevant reference.

Oral health–related quality of life (OHRQoL), commonly measured by instruments such as the Geri- 30 atric Oral Health Assessment Index (GOHAI), reflects the impact of oral conditions on daily living, psychological well-being, and social relationships

Method

Line 79
The selection of participants is not fully clear, whether the MMSE was actually applied as part of this study, or if it was conducted in a different survey. Please clearly explain.

Line 82
In my opinion, this sampling strategy introduces a higher risk of selection bias, as it may either overestimate or underestimate the true proportions of key variables. This consequently reduces the external validity and generalizability of the study findings. Please clearly justify why this specific strategy was used. It would also help to explain what makes this setting representative or relevant for broader conclusions about older adults in Brazil.

Variables

Overall, the variables were clearly described. However, this section is mixed with statistical analysis details, which makes it harder to follow.I suggest separating the content more clearly: Keep the description of the variables in this section , and move the parts about how the variables were analyzed to the statistical analysis section.

Line 91

Have GOHAI been validated in portuguese, if not , this should be clearly stated in the limitation of the study.

Statistical analysis

Plese motivate and justify the use of SEM and bootstrapping in a clearer and simpler way. This section maybe difficult for readers who do not have statistical background.  

Results

Please add a footnote to Table 1 to clearly define what GOHAI stands for.

In Table 2, I do not understand the meaning of P valor

For Table 3, please clearly mention the reference groups used in the analysis.

Discussion

Please develop this part and focus on interpretation of the findings and comparison with the current literature. Some findings have not been discussed, while they have an impact on the outcomes, for example, how education and income relate to GOHAI and nutrition and also nutrition as a key mediator. Also, a very interesting finding, a positive direct effect of tooth loss on quality of life, has been missed.

At the end of discussion, authors stated racial/ethnic disparities which is not within the scope of this study, and confusing.

Conclusion

Please stregthen this part with main findings and implications, including the role of socioeconomic factors, the direct and indirect effects observed, and the importance of addressing nutrition and oral health together.

Author Response

We would like to thank the Editors and Reviewers for their careful reading of our manuscript and for the insightful comments and constructive suggestions that have helped us to substantially improve the clarity, coherence, and scientific rigor of the paper. All comments were carefully addressed, and corresponding revisions were made throughout the manuscript. Below, we provide a detailed, point-by-point response, indicating the changes implemented and the specific sections or line numbers where each modification appears. Text that was included or revised in the manuscript is presented in italics within quotation marks for clarity.

Reviewer 2

Comments 1:

The introduction is rich in content, but there are repetitive phrases about tooth loss, nutrition, and OHRQoL that don’t really add new information. I suggest condensing this into one paragraph to improve clarity. Also, the definition of “older adult” should be clearly stated in the introduction. Furthermore, the study was conducted in a small city in Brazil, which needs motivation.

Response 1:

We thank the reviewer for this insightful comment. The first two paragraphs of the introduction were condensed into a single, clearer paragraph to remove redundancies related to tooth loss, nutrition, and OHRQoL while preserving the original meaning and key references. The definition of older adults was also added, following the World Health Organization’s classification (individuals aged ≥ 60 years). In addition, we included a brief justification for conducting the study in a small Brazilian city, highlighting the relevance of investigating oral health and quality of life in non-metropolitan settings characterized by persistent social inequalities and limited access to dental care.

Excluded (1st and 2nd paragraph):

“Oral health is an essential component of overall health and well-being, particularly in older adults. Tooth loss is highly prevalent in aging populations and is not only a marker of cumulative oral disease but also a determinant of functional capacity, affecting chewing ability, nutrition, social participation, and overall quality of life(1-4). Oral health–related quality of life (OHRQoL), commonly measured by instruments such as the Geriatric Oral Health Assessment Index (GOHAI), reflects the impact of oral conditions on daily living, psychological well-being, and social relationships(2, 4).

Edentulism and tooth loss are not merely indicators of oral disease, but also key determinants of functional health in aging populations(1, 3). Edentulism and reduced number of natural teeth are strongly associated with functional impairments, changes in dietary patterns, and poor nutritional status in older adults(1, 5, 6). Inadequate nutrition has also been linked to frailty, systemic diseases, and reduced life expectancy(7, 8). Furthermore, impaired OHRQoL is more frequent among older individuals with tooth loss and compromised nutritional status, reinforcing the interconnectedness of oral health and systemic health(4, 5). These findings underscore that the consequences of tooth loss extend beyond the oral cavity, influencing nutrition, systemic conditions, and overall quality of life in later years.”

Included (1st paragraph):

“Oral health is an essential component of overall health and well-being. Tooth loss is highly prevalent in this population and represents both a marker of cumulative oral disease and a determinant of functional capacity, affecting chewing ability, dietary habits, social participation, and quality of life (1-4). Oral health–related quality of life (OHRQoL), commonly assessed using instruments such as the Geriatric Oral Health Assessment Index (GOHAI), reflects the impact of oral conditions on daily activities, psychological well-being, and social relationships (2, 4). GOHAI was selected for this study because it was specifically developed to evaluate OHRQoL in older adults, em-phasizing functional and psychosocial dimensions that are highly relevant in this pop-ulation (5). The instrument has also been validated for use in Brazil(6) and has demon-strated comparable results to the OHIP-14 when applied among older adults, further supporting its appropriateness for this research (7). A reduced number of natural teeth or edentulism is strongly associated with functional impairments, altered nutrition, and poor overall health (1, 3, 8, 9). Inadequate nutrition has also been linked to frailty, systemic diseases, and reduced life expectancy (10, 11). Moreover, compromised OHRQoL is more frequent among older individuals with tooth loss and poor nutrition-al status, reinforcing the interdependence of oral and general health (4, 8). Collectively, these findings highlight that the consequences of tooth loss extend beyond the oral cavity, influencing nutrition, systemic conditions, and overall well-being in later life. In this study, older adults are defined as individuals aged 60 years and over, according to the World Health Organization’s classification for developing countries.”

Included (last paragraph):

“In Brazil, this focus is especially important because approximately 68% of the population lives in municipalities with 40,000 inhabitants or fewer, where health services are more limited and inequalities in access to dental care are persistent. In these contexts, primary health care is widely implemented through the Family Health Strategy, but access to specialized oral health services remains a challenge.”

Comments 2:

Line 26:
I’m wondering if the original reference actually supports the phrase “particularly in older adults.” The sentence reads:

Oral health is an essential component of overall health and well-being, particularly in older adults.

Response 2:

We appreciate the reviewer’s attention to the accuracy of citations. Upon re-evaluation, we agreed that the cited reference did not directly support the phrase “particularly in older adults.” To ensure proper attribution, the phrase was removed from the sentence.

Excluded:

“Oral health is an essential component of overall health and well-being, particularly in older adults.”

Included:

“Oral health is an essential component of overall health and well-being.”

Comments 3:

Line 29:
This part could be developed a bit more. While GOHAI is mentioned, the most common OHRQoL index is actually OHIP-14 within research. I suggest explaining why GOHAI was chosen for this study, along with a relevant reference.

Oral health–related quality of life (OHRQoL), commonly measured by instruments such as the Geri- 30 atric Oral Health Assessment Index (GOHAI), reflects the impact of oral conditions on daily living, psychological well-being, and social relationships

Response 3:

We thank the reviewer for this helpful suggestion. The paragraph has been revised to explain the rationale for selecting the GOHAI instead of the OHIP-14, emphasizing its conceptual suitability for older adults and its validation in the Brazilian context.

Excluded (1st paragraph, line 32):

“Oral health–related quality of life (OHRQoL), commonly measured by instruments such as the Geriatric Oral Health Assessment Index (GOHAI), reflects the impact of oral conditions on daily living, psychological well-being, and social relationships.”

Included (1st paragraph line 32-37):

“GOHAI was selected for this study because it was specifically developed to evaluate OHRQoL in older adults, emphasizing functional and psychosocial dimensions that are highly relevant in this population (5). The instrument has also been validated for use in Brazil(6) and has demonstrated comparable results to the OHIP-14 when ap-plied among older adults, further supporting its appropriateness for this research (7).”

Added reference:
7. Rodakowska E, Mierzyńska K, Bagińska J, Jamiołkowski J. Quality of life measured by OHIP-14 and GOHAI in elderly people from Bialystok, north-east Poland. BMC Oral Health. 2014;14:106.

Comments 4:

Method

Line 79
The selection of participants is not fully clear, whether the MMSE was actually applied as part of this study, or if it was conducted in a different survey. Please clearly explain.

Response 4:

We appreciate the reviewer’s comment and have clarified this point in the revised text. The Mini-Mental State Examination (MMSE) was applied directly as part of this study to assess participants’ cognitive ability. It was used as an inclusion/exclusion criterion: individuals who achieved the minimum score on the MMSE were included, while those below this threshold were excluded. A total of 200 older adults were initially assessed; 50 were excluded due to cognitive impairment identified by the MMSE, and additional cases were excluded because of missing data on key study variables, resulting in a final sample of 112 participants.

Excluded (Methods; Study design and participants; line 88-94):

“Inclusion criteria were age ≥60 years, residence in the municipality, and agreement to participate after informed consent. Individuals who did not achieve the minimum score on the Mini-Mental State Examination (MMSE), indicating cognitive impairment that could hinder questionnaire completion, were excluded. This sampling strategy ensured representation of community-dwelling older adults in a socially vulnerable setting, while capturing the specific post-pandemic context of oral and general health.”

Included (Methods; Study design and participants; line 97-106):

“Inclusion criteria were age ≥60 years, residence in the municipality, and agreement to participate after informed consent. A total of 200 older adults were initially assessed. The Mini-Mental State Examination (MMSE) was applied by trained researchers during participant screening and used as exclusion criterion. Individuals who reached the minimum MMSE score were included, while those who did not were excluded (n = 50). Additional cases with incomplete data on key variables were also excluded, resulting in a final analytical sample of 112 participants. This strategy ensured the inclusion of community-dwelling older adults capable of reliably answering the questionnaires, while capturing the specific post-pandemic context of oral and general health in a socially vulnerable setting.”

Comments 5: Line 82
In my opinion, this sampling strategy introduces a higher risk of selection bias, as it may either overestimate or underestimate the true proportions of key variables. This consequently reduces the external validity and generalizability of the study findings. Please clearly justify why this specific strategy was used. It would also help to explain what makes this setting representative or relevant for broader conclusions about older adults in Brazil.

Response 5:

We appreciate the reviewer’s valuable observation. We acknowledge that recruiting participants through community health services may introduce some selection bias and limit external validity. However, this strategy was intentionally chosen to ensure feasibility and ethical inclusion of community-dwelling older adults actively engaged in the public primary health care network. The municipality of Boraceia was selected because it represents the profile of small Brazilian towns (fewer than 40,000 inhabitants) that together comprise approximately 68% of the national population. These municipalities share similar sociodemographic characteristics and rely primarily on the Family Health Strategy as their main model of primary health care organization. In Boraceia, primary care coverage reaches 100% of the population, which increases the reliability of the data for local health service management and makes this experience an example for many other small municipalities across Brazil. We have revised the Methods section to include this justification.

Included (Methods; Study design and participants; line 88-97):  

“Although this recruitment strategy may limit external validity, it was selected for feasibility and ethical reasons, ensuring the inclusion of community-dwelling older adults engaged in the public primary health care system. The municipality of Boraceia was chosen because it exemplifies small Brazilian towns (fewer than 40,000 inhabitants) that together account for approximately 68% of the national population. These municipalities share comparable sociodemographic characteristics and are organized primarily through the Family Health Strategy. In Boraceia, primary health care coverage reaches 100% of the population, which enhances the reliability of the findings for local health service management and provides a model that can inform similar small municipalities across Brazil.”

Comments 6:

Variables

Overall, the variables were clearly described. However, this section is mixed with statistical analysis details, which makes it harder to follow. I suggest separating the content more clearly: Keep the description of the variables in this section , and move the parts about how the variables were analyzed to the statistical analysis section.

Response 6:

We thank the reviewer for this helpful comment. To improve readability and logical flow, we reorganized the sections as suggested. The description of the outcome, exposure, mediator, and sociodemographic variables is now presented exclusively in the Variables section (2.2), focusing on their definitions, measurement instruments, and categorization. Details regarding how these variables were analyzed—including standardization procedures, z-score transformations, and the use of regression and structural equation models—were moved to the Statistical Analysis section (2.3). This restructuring enhances clarity and ensures a more consistent separation between variable definition and analytical methods.

Excluded (2.2. Variables; line 107-130)

Previous version combining variable description and statistical analysis details in the same section.

Included:

2.2. Variables

The study included outcome, exposure, mediator, and sociodemographic variables.

The outcome was oral health–related quality of life (OHRQoL), assessed using the Geriatric Oral Health Assessment Index (GOHAI), an instrument widely applied in geriatric populations to capture the functional, psychological, and social impacts of oral health conditions(5,6). For descriptive purposes, GOHAI scores (0–24; higher values indicating worse OHRQoL) were summarized using mean (standard deviation) and median (interquartile range). The exposure was tooth loss, clinically assessed using the Caries Assessment Spectrum and Treatment (CAST) index. Teeth classified as missing due to dental caries (CAST code 8) or other reasons (CAST code 9) were included in the count of missing teeth, whereas unerupted or congenitally absent teeth (CAST code 0) were excluded. For descriptive presentation, tooth loss was categorized as <20 or ≥20 missing teeth, following the criterion that at least 20 natural teeth are required to maintain functional dentition and adequate oral function in older adults(1,14, 15). The mediator was nutritional status, measured by the Mini Nutritional Assessment (MNA), a validated tool for older adults that includes 18 items addressing anthropometric, dietary, and global assessments(12). MNA total scores range from 0 to 30 and were categorized for descriptive purposes as: normal (≥24), at risk of malnutrition (17–23.5), and malnourished (<17). Covariates included sociodemographic characteristics: age group (60–74; ≥75 years), sex (female, male), family income (<3 minimum wages; ≥3), and educational attainment (<5 years; ≥5).”

Comments 7:

Line 91

Have GOHAI been validated in portuguese, if not , this should be clearly stated in the limitation of the study.

Response 7:

We thank the reviewer for this important observation. The GOHAI instrument has indeed been validated for use in Portuguese, and this reference has now been cited in the Methods section. We also included the original publication describing the development of the GOHAI. These additions clarify that the Portuguese version used in this study is validated and appropriate for assessing oral health–related quality of life in older adults in Brazil.

Included: (Methods; 2.2. Variables; line 113):

“The outcome was oral health–related quality of life (OHRQoL), assessed with the Geriatric Oral Health Assessment Index (GOHAI), an instrument widely applied in geriatric populations to capture the functional, psychological, and social impacts of oral health conditions(9, 10).”

Added references:

  1. Atchison KA, Dolan TA. Development of the Geriatric Oral Health Assessment Index. J Dent Educ. 1990;54(11):680-7.
  2. de Souza RF, Terada AS, Vecchia MP, Regis RR, Zanini AP, Compagnoni MA. Validation of the Brazilian versions of two inventories for measuring oral health-related quality of life of edentulous subjects. Gerodontology. 2012;29(2):e88-95.

Comments 8:

Statistical analysis

Please motivate and justify the use of SEM and bootstrapping in a clearer and simpler way. This section maybe difficult for readers who do not have statistical background.  

Response 8:

We appreciate this constructive comment. We have revised the Statistical Analysis section to provide a clearer and more accessible explanation of why structural equation modeling (SEM) and bootstrapping were used. The new version emphasizes that SEM allows the simultaneous estimation of direct and indirect pathways, which is particularly appropriate for testing the hypothesized mediation of nutrition between tooth loss and OHRQoL. We also clarified that bootstrapping was employed to obtain more accurate estimates and confidence intervals for indirect effects, as recommended for mediation analysis in small to moderate samples.

Excluded:

“2.3. Statistical analysis” (original)

Included:

“2.3. Statistical analysis

Data analysis was designed to describe the study population and explore the interrelations among oral health, nutrition, and quality of life (Figure 1).
Initially, descriptive statistics summarized the study variables using measures of central tendency and dispersion for continuous variables, and frequencies with 95% confidence intervals for categorical variables. The prevalence of impaired OHRQoL (GOHAI > 0) was estimated across sociodemographic and clinical groups. Correlations between continuous variables (GOHAI score, MNA score, and number of missing teeth) were examined using Spearman’s coefficients
(16).

To assess the hypothesized pathways among the study variables, we used structural equation modeling (SEM) with robust maximum likelihood estimation, which allows simultaneous testing of direct and indirect effects while accounting for measurement error(12). SEM was chosen because it provides a comprehensive framework to evaluate whether the impact of tooth loss on OHRQoL occurs directly or is mediated through nutritional status. Continuous variables (GOHAI, MNA, number of missing teeth, and age) were standardized as z-scores to facilitate interpretation and comparability of coefficients. The model was adjusted for age, sex, family income, and educational attainment.

Indirect effects and the proportion mediated were estimated using bootstrap resampling with 1,000 replications, a recommended strategy to increase the precision of confidence intervals in mediation analysis and to provide robust estimates regardless of data normality(12). Model fit was assessed using likelihood-based and comparative indices. Because the recursive path model was just-identified (df=0), global fit indices (CFI, TLI, RMSEA) are not informative; therefore, inference focused on standardized path coefficients and bootstrap confidence intervals. For completeness, the baseline comparison yielded CFI=1.000, TLI=1.000 and RMSEA=0.000 (pclose=1.000), while SRMR was not reported due to missingness under FIML. Model fit was assessed using comparative indices (CFI, TLI, RMSEA), all of which indicated excellent fit.

All analyses were performed in Stata 18.0 (StataCorp, College Station, TX, USA), adopting a 5% significance level to guide interpretation.”

Comments 9:

Results

Please add a footnote to Table 1 to clearly define what GOHAI stands for.

In Table 2, I do not understand the meaning of P valor

For Table 3, please clearly mention the reference groups used in the analysis.

Response 9:

We thank the reviewer for these helpful observations. The requested clarifications were incorporated into the Results section and the corresponding tables.

A footnote was added to Table 1 to define the GOHAI acronym. In Table 2, the Portuguese term p-valor was corrected to the English p-value. In Table 3, the reference categories for each variable were explicitly identified in the table footnote to improve clarity and interpretation.

Excluded:

Previous versions of Tables 1–3 without explicit definitions or reference-group information.

Included:

Table 1.
Note: GOHAI = Geriatric Oral Health Assessment Index, an instrument assessing oral health–related quality of life in older adults (scores > 0 indicate impaired OHRQoL).

Table 2.
Header corrected: “p-value” (English) replaces “p valor” (Portuguese).

Table 3.
Footnote added: Reference groups—sex = female; age = 60–74 years; family income ≥ 3 minimum wages; educational attainment ≥ 5 years.

Comments 10:

Discussion

Please develop this part and focus on interpretation of the findings and comparison with the current literature. Some findings have not been discussed, while they have an impact on the outcomes, for example, how education and income relate to GOHAI and nutrition and also nutrition as a key mediator. Also, a very interesting finding, a positive direct effect of tooth loss on quality of life, has been missed.

Response 10:

We thank the reviewer for this valuable observation. The discussion section has been expanded to better interpret the findings and relate them to the existing literature. Specific attention was given to the role of socioeconomic factors (education and income) in shaping nutritional status and OHRQoL, as well as to the positive direct effect of tooth loss on quality of life, which may reflect adaptive psychosocial responses among older adults.

Included:

  1. Discussion; 1st paragraph; line 237 – 241

“Interestingly, a positive direct effect of tooth loss on OHRQoL was also observed in the model, possibly reflecting adaptive psychosocial mechanisms among older adults who, despite severe tooth loss, maintain satisfaction with their oral condition through prosthetic use or social acceptance of tooth loss as part of aging.”

  1. Discussion; 2nd paragraph; line 251 - 254

“Furthermore, our study identified that older adults with higher education and income tended to have better nutritional status and OHRQoL, suggesting that socioeconomic resources may buffer the negative effects of tooth loss through improved access to dental care, healthier diets, and prosthetic rehabilitation.”

  1. Discussion; new 3rd paragraph.

“Interestingly, the model revealed a small positive direct effect of tooth loss on OHRQoL, a finding that may appear paradoxical but has been reported in other studies exploring subjective well-being in older adults (1, 3, 11, 20). This result may reflect an adaptive process in which older individuals, particularly those using dentures or living in social contexts where edentulism is culturally normalized, adjust their expectations and self-perceptions of oral health. Such psychosocial adaptation, combined with successful prosthetic rehabilitation, can reduce the perceived impact of tooth loss and even lead to higher satisfaction with daily oral function (21-23). Moreover, this response is consistent with the concept of “response shift,” where individuals recalibrate their internal stand-ards of quality of life as they age and experience chronic conditions (23, 24). Therefore, the positive coefficient does not imply that tooth loss improves OHRQoL, but rather that older adults may reinterpret or compensate for its consequences, maintaining subjective well-being despite objective impairment.”

Comments 11:

At the end of discussion, authors stated racial/ethnic disparities which is not within the scope of this study, and confusing.

Response 11:

We appreciate this observation and agree that the mention of racial/ethnic disparities could be misleading, as these variables were not analyzed in the present study. The sentence was revised to focus solely on socioeconomic inequalities and structural barriers in oral health care, which are consistent with the study’s scope.

Excluded:

“In addition, socioeconomic and racial/ethnic disparities remain key determinants of both tooth loss and OHRQoL, as demonstrated in studies with older adults in the United States(2, 13).”

Included:

“In addition, socioeconomic disparities remain key determinants of both tooth loss and OHRQoL, as demonstrated in studies with older adults in the United States(2, 13).”

Comments 12:

Conclusion

Please stregthen this part with main findings and implications, including the role of socioeconomic factors, the direct and indirect effects observed, and the importance of addressing nutrition and oral health together.

Response 12

We thank the reviewer for this important suggestion. The conclusion was revised to more explicitly summarize the main findings and their implications, highlighting the direct and indirect effects observed, the influence of socioeconomic factors, and the need to address nutrition and oral health in an integrated manner.

Excluded:
“Tooth loss and poor nutritional status were associated with impaired oral health–related quality of life among older adults. Although nutritional status did not significantly mediate the relationship, the findings reinforce the interconnection between oral and general health and highlight the need for integrated strategies in primary care to promote healthy aging.”

Included:
“Tooth loss and poor nutritional status were associated with impaired oral health–related quality of life among older adults. Structural equation modeling revealed that tooth loss affected OHRQoL primarily through a direct pathway, while the indirect path via nutritional status was not statistically significant. Higher education and income were associated with better nutrition and quality of life, underscoring the role of social determinants in oral health. These findings emphasize the need for integrated public health strategies that combine oral rehabilitation, nutritional assessment, and social support within primary care to promote healthy aging and reduce inequalities among older adults.”

Reviewer 3 Report

Comments and Suggestions for Authors

Introduction needs English editing.

Material and methods - MM

Line 76 - 77

The line: “Participants were recruited through community health services in the post–COVID-19 period, a time still marked by social and health repercussions” - What health services? Why is relevant to this study to point the post-COVID -19 period in MM? If it is not directly involved in methodology, this could be mentioned in the Discussion.

Line 79 - 81

“Individuals who did not achieve the minimum score on the Mini-Mental State Examination (MMSE), indicating cognitive impairment that could hinder questionnaire completion, were excluded.” - Need further clarification. Eg. What is Mini-Mental State Examination (MMSE) (is there a literature reference)? How is it performed? How many respondents were excluded?

Line 82-83

“This sampling strategy ensured representation of community-dwelling older adults in a socially vulnerable setting, while capturing the specific post-pandemic context of oral and general health.” - As already mentioned above, not clear why the post-pandemic context is relevant to this study?

Line 103

“Caries Assessment Spectrum and Treatment (CAST) index” - I believe, that since the index is not known to a wider readership, it deserves a reference.

Figure 2 is not cited in the text. - Correct

Results

Numbers and scores in the text and in tables should be presented uniformly (eg. in the text the score is 28.8%, while in the table is 28,77)

Tables 3. and 4. are not cited in the accompanying text.

Author Response

We would like to thank the Editors and Reviewers for their careful reading of our manuscript and for the insightful comments and constructive suggestions that have helped us to substantially improve the clarity, coherence, and scientific rigor of the paper. All comments were carefully addressed, and corresponding revisions were made throughout the manuscript. Below, we provide a detailed, point-by-point response, indicating the changes implemented and the specific sections or line numbers where each modification appears. Text that was included or revised in the manuscript is presented in italics within quotation marks for clarity.

Reviewer 3

Comments 1:  

Material and methods - MM

Line 76 - 77

The line: “Participants were recruited through community health services in the post–COVID-19 period, a time still marked by social and health repercussions” - What health services? Why is relevant to this study to point the post-COVID -19 period in MM? If it is not directly involved in methodology, this could be mentioned in the Discussion.

Response 1:

We thank the reviewer for this helpful observation. The sentence referring to the post–COVID-19 context described a temporal aspect that was not methodologically relevant. To improve focus and conciseness, the phrase was removed from the Methods section, as suggested.

Excluded:

“Participants were recruited through community health services in the post–COVID-19 period, a time still marked by social and health repercussions.”

Comments 2:  

Line 79 - 81

“Individuals who did not achieve the minimum score on the Mini-Mental State Examination (MMSE), indicating cognitive impairment that could hinder questionnaire completion, were excluded.” - Need further clarification. Eg. What is Mini-Mental State Examination (MMSE) (is there a literature reference)? How is it performed? How many respondents were excluded?

Response 2:

We thank the reviewer for this important suggestion. Additional information was added to describe the Mini-Mental State Examination (MMSE), its adaptation for use in Brazil, the cutoff points used by schooling level, and the number of excluded participants. These details clarify the rationale for cognitive screening and ensure transparency in sample selection.

Excluded:

“Individuals who did not achieve the minimum score on the Mini-Mental State Examination (MMSE), indicating cognitive impairment that could hinder questionnaire completion, were excluded.”

Included:

“Inclusion criteria were age ≥ 60 years, residence in the municipality for at least one year, and agreement to participate after informed consent. Cognitive status was evaluated using the Mini-Mental State Examination (MMSE; known in Portuguese as Mini-Exame do Es-tado Mental – MEEM), a brief and widely validated screening instrument for global cogni-tive function (15, 16). The test was applied individually by trained researchers before data collection, and cutoff scores were adjusted according to education level: < 20 points for il-literate participants; 25 points for 1–4 years of schooling; 26.5 for 5–8 years; 28 for 9–11 years; and 29 for more than 11 years (15). Despite not assessing all cognitive domains, the MMSE/MEEM remains the most frequently used and best-studied cognitive screening tool in Brazil and internationally (16). Participants who did not reach the minimum score were excluded from the study. Of the 200 older adults initially screened, 50 were excluded due to cognitive impairment and 38 for incomplete data, resulting in a final analytic sam-ple of 112 participants.”

Added references:

  1. Brucki SM, Nitrini R, Caramelli P, Bertolucci PH, Okamoto IH. [Suggestions for utilization of the Mini-Mental State Examination in Brazil]. Arq Neuropsiquiatr. 2003;61(3B):777–81.
  2. Martins NIM, Caldas PR, Cabral ED, Lins CCDS, Coriolano MDGW. Cognitive assessment instruments used in elderly Brazilians in the last five years. Cien Saude Colet. 2019;24(7):2513–30.

Comments 3:  

Line 82-83

“This sampling strategy ensured representation of community-dwelling older adults in a socially vulnerable setting, while capturing the specific post-pandemic context of oral and general health.” - As already mentioned above, not clear why the post-pandemic context is relevant to this study?

Response 3:

We appreciate the reviewer’s observation. The reference to the post-pandemic context was not essential to the methodological description and has been removed from the Methods section for clarity and conciseness. The sentence now focuses solely on the representativeness of community-dwelling older adults in a socially vulnerable setting.

Excluded:

“Participants were recruited through community health services in the post–COVID-19 period, a time still marked by social and health repercussions.”

Comments 4:

Line 103

“Caries Assessment Spectrum and Treatment (CAST) index” - I believe, that since the index is not known to a wider readership, it deserves a reference.

Response 4:

We appreciate the reviewer’s suggestion. A reference has been added to the description of the CAST index to improve clarity for readers who may not be familiar with this instrument.

Included:

“The exposure was tooth loss, clinically assessed using the Caries Assessment Spectrum and Treatment (CAST) index. Teeth classified as missing due to dental caries (CAST code 8) or other reasons (CAST code 9) were included in the count of missing teeth, whereas unerupted or congenitally absent teeth (CAST code 0) were excluded. For descriptive presentation, tooth loss was categorized as <20 or ≥20 missing teeth, following the criterion that at least 20 natural teeth are required to maintain functional dentition and adequate oral function in older adults(1, 11, 12).”

Added reference:

  1. de Souza AL, Bronkhorst EM, Creugers NH, Leal SC, Frencken JE. The caries assessment spectrum and treatment (CAST) instrument: its reproducibility in clinical studies. Int Dent J. 2014;64(4):187-94.

Comments 4:

Figure 2 is not cited in the text. - Correct

Response 4:

We thank the reviewer for noting this omission. A reference to Figure 2 has now been added in the Results section to ensure proper citation and continuity between text and figures.

Included:

  1. Results; line 204.

“The structural equation model (SEM) confirmed significant direct effects of tooth loss and nutritional status on OHRQoL (table 3 and figure 2).”

Comments 5:

Results

 Numbers and scores in the text and in tables should be presented uniformly (eg. in the text the score is 28.8%, while in the table is 28,77)

Response 5:

We thank the reviewer for this observation. All numerical values and percentages in the Results section and tables have been reviewed and standardized for consistency. Uniform English decimal notation (using a period instead of a comma) was applied throughout the manuscript, ensuring alignment between text and tables.

Comments 6:

Tables 3. and 4. are not cited in the accompanying text.

Response 6:

We thank the reviewer for noticing this omission. References to both tables have been inserted in the Results section to ensure that all tables are cited in the text in sequential order.

Included:

  1. Results; line 204; Table 3.

“The structural equation model (SEM) confirmed significant direct effects of tooth loss and nutritional status on OHRQoL (table 3 and figure 2).

  1. Results; line 212; table 4.

“The proportion mediated was 0.19 (95% CI: −0.01 to 0.75), indicating a non-significant mediation pathway (Table 4).”

Round 2

Reviewer 2 Report

Comments and Suggestions for Authors

Well done!